# Static and Sequential Malicious Attacks in the Context of Selective Forgetting

**Chenxu Zhao**[*]
Department of Computer Science
Iowa State University
cxzhao@iastate.edu

**Wei Qian**[*]
Department of Computer Science
Iowa State University
wqi@iastate.edu

**Rex Ying**
Department of Computer Science
Yale University
rex.ying@yale.edu

**Mengdi Huai**
Department of Computer Science
Iowa State University
mdhuai@iastate.edu

## Abstract

With the growing demand for the right to be forgotten, there is an increasing need for machine learning models to forget sensitive data and its impact. To address this, the paradigm of selective forgetting (a.k.a machine unlearning) has been extensively studied, which aims to remove the impact of requested data from a well-trained model without retraining from scratch. Despite its significant success, limited attention has been given to the security vulnerabilities of the unlearning system concerning malicious data update requests. Motivated by this, in this paper, we explore the possibility and feasibility of malicious data update requests during the unlearning process. Specifically, we first propose a new class of malicious selective forgetting attacks, which involves a static scenario where all the malicious data update requests are provided by the adversary at once. Additionally, considering the sequential setting where the data update requests arrive sequentially, we also design a novel framework for sequential forgetting attacks, which is formulated as a stochastic optimal control problem. We also propose novel optimization algorithms that can find the effective malicious data update requests. We perform theoretical analyses for the proposed selective forgetting attacks, and extensive experimental results validate the effectiveness of our proposed selective forgetting attacks. *The source code is available in the supplementary material.*

## 1 Introduction

Machine learning algorithms play a crucial role in diverse fields such as biology, speech recognition, agriculture, and medicine. To build pertinent models, these algorithms are frequently trained using a range of data sources, including third-party datasets, internal datasets, and customized subsets of publicly available user data. With recent demands for increased data privacy, the data users could erase the impact of their sensitive information from the trained models to ensure their privacy. Recent legislation (e.g., the General Data Protection Regulation from the European Union [60], the California Consumer Privacy Act [47], and the Canada's proposed Consumer Privacy Protection Act) requires the right to be forgotten, and grants users an unconditional right to request that their private data be removed from everywhere in the system within a reasonable time.

---

[*]The first two authors contribute equally to this work.

37th Conference on Neural Information Processing Systems (NeurIPS 2023).

However, with the development of traditional machine learning techniques, this basic right is usually neglected or violated [70, 19, 38]. An illustrative instance is the inadvertent leakage of patients' genetic markers through machine learning methods employed for genetic data processing without the patients' awareness. Therefore, it is important to entitle data users the right to delete their personal data from trained machine learning models since machine learning models could memorize sensitive information of the training data and thus expose individual's privacy risk [54, 6, 8, 30, 59, 4, 40]. The most naive way is to retrain from the original data after removing the samples that need to be forgotten. Unfortunately, this naive retraining method can be prohibitive in terms of the computational and space cost—especially for large models and frequent deletion requests. To mitigate this, selective forgetting (a.k.a machine unlearning) [4, 9, 43, 22, 20, 28, 63] has been extensively researched in recent years to avoid the high computational cost associated with fully retraining a model from scratch.

However, existing studies on selective forgetting mainly focus on designing new forgetting algorithms to enable the right to be forgotten to be efficiently implemented, leaving the security issues during the unlearning process in adversarial settings largely unexplored. In practice, the motivated adversary could make use of the unlearning pipeline to craft malicious data update requests to achieve his/her desired attack goals. For example, the motivated adversary could increase a disadvantage against a specific group of individuals. In Figure 1, we present a toy example to highlight the impact of malicious data update requests on fairness using the COMPAS [32] dataset. Note that fairness in machine learning [67, 55] refers to the concept of ensuring that the decisions produced by machine learning algorithms are unbiased and equitable across different groups of individuals,

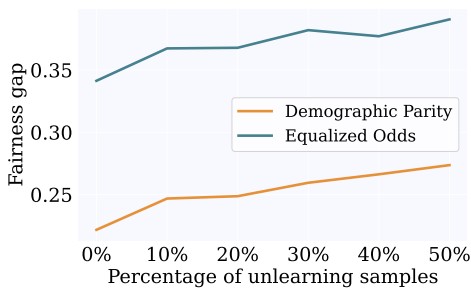

Figure 1: Fairness gap (Demographic Parity and Equalized Odds) with random unlearning samples in the minority group on COMPAS.

irrespective of their protected attributes (e.g., race and ethnicity). Here we consider race (black/white) as the sensitive feature and just randomly unlearn varying percentages of samples from the minority group (white). For the fairness evaluation metrics, we adopt Demographic Parity [16] and Equalized Odds [24] (please refer to the supplementary material for details of the two evaluation metrics and more experimental results on attacking fairness). From this figure, we can easily see that even though we just randomly delete some individuals from the white group in the dataset, the fairness gap expands, indicating that the model's fairness is compromised. When the victim systems are employed for security-sensitive applications, such malicious data update requests can cause tremendous security threats to the unlearning system. Therefore, it is essential to understand the feasibility of malicious data requests in the unlearning systems.

In this paper, we aim to conduct a comprehensive study on the security vulnerability and robustness of the unlearning system to malicious data update requests during the unlearning process. Specifically, we first propose a novel static selective forgetting attack framework, where the adversary exploits vulnerabilities in the unlearning systems by submitting a set of carefully crafted data update requests at once. More specifically, the proposed static attack framework uses discrete indication variables to formulate the complete deletion of targeted training samples, which is very hard to be directly solved. To address this challenge, we design a continuous and differentiable function to approximate the discrete component. On the other hand, many real-world applications involve streaming data update requests that arrive in a sequential manner. The adversary could take advantage of this sequential interaction setting to strategically manipulate the unlearning process. However, launching attacks on all the received sequential data update requests indiscriminately can potentially lead to the detection of the adversary. To address this issue, we also design a novel sequential selective forgetting attack framework that takes into account the order and timing of data update requests. In this framework, the adversary focuses on attacking a few critical data update requests to maximize the impact of his/her malicious actions, potentially leading to severe security threats. We also conduct theoretical analyses of our proposed selective forgetting attacks. Further, we conduct extensive experiments in different scenarios to validate the general effectiveness of our proposed selective forgetting attacks.

## 2    Related Work

Selective forgetting (a.k.a machine unlearning and data deletion ) [20, 38, 19, 38, 49, 62] refers to removing the influence of the requested data from a trained model. However, existing data deletion works [19, 19, 8, 30, 59, 4, 10, 9, 43, 22, 20, 28, 63] ignore the risks of malicious data update requests during the unlearning process and fail to identify the vulnerabilities of machine learning models to selective forgetting attacks. It is worth mentioning that our proposed selective forgetting attacks are different from traditional evasion attacks [66, 68, 3, 34, 61, 27, 69, 26] and data poisoning attacks [46, 14, 51, 39]. Evasion attacks achieve adversarial goals by modifying test samples; however, our forgetting attacks do not change test data and only modify the well-trained models by making malicious data update requests. Traditional data poisoning attacks occur during training, and manipulate the original clean training data. Note that [13] adds carefully crafted samples to the training dataset, and assumes the exact and retraining tasks. The authors in [48] fail to address the issue of malicious whole data deletion in the static setting, nor do they consider the sequential attack setting involving various types of malicious data update requests.

## 3    Malicious Selective Forgetting Attacks

Without loss of generality, in this work, we consider the classification models. Let $D = \{z_i = (x_i, y_i)\}_{i=1}^N$ denote the training dataset from $\mathcal{Z} = \mathcal{X} \times \mathcal{Y}$, where $x_i \in \mathbb{R}^{d_1}$ is a $d_1$-dimensional sample and $y_i \in [C]$ denotes its associated class label. The model owner applies a learning algorithm $A$ on $D$ to learn a model $f(\theta)$ parameterized by $\theta$, such that $f(\theta)$ achieves low empirical loss. We denote a loss function by a mapping $\ell : \Theta \times \mathcal{Z} \to \mathbb{R}$ that takes the parameters $\theta \in \Theta$ and a sample $z \in \mathcal{Z}$, outputs the loss of $\theta$ on $z$ (i.e., $\ell(z; \theta)$ that we denote by $\ell_z(\theta)$). Let $A(D)$ denote the distribution over the resulting model parameters $\theta$ when the learning algorithm $A$ is applied on $D$.

**Threat Model.** As previously mentioned, the proposed malicious selective forgetting attacks are unlearning time attacks, i.e., the adversary interferes with the unlearning process of the well-trained model and cannot modify any test sample submitted to the victim model at testing time. In addition, the adversary is unable to modify the training samples during the training stage. This reflects the unlearning scenario in which the adversary can only generate the update requests to selectively forget certain data information during the unlearning process. In this paper, we study both the *white-box* and *black-box* settings. Specifically, in the white-box setting, we make the assumption that the adversary possesses complete information about the system (including the model architecture and parameters of the well-trained model). Note that many evasion and poisoning attacks in the literature [5, 42, 65, 52, 18, 33, 29] employ a white-box model to study the adversary's strong attack behaviors in such worst-case settings. In the black-box setting, we assume that the adversary does not have any prior knowledge about the target well-trained model.

### 3.1    Static Selective Forgetting Attacks

Here, we consider the static attack setting, where the model holder owns a well-trained classification model $f(\theta)$ parameterized by $\theta \in \mathbb{R}^{d_2}$ on dataset $D$, and the adversary aims to make malicious update requests to deliberately forget some information to achieve his/her desired attack goals. In this case, all the malicious data update requests are provided at once, without consideration for the sequential order. We use $D_f \subset D$ and $R_A$ to denote the adversary's requested forget set and the coupled unlearning algorithm for $A$, respectively. Note that for the given forget set, machine unlearning method can return a model $\theta \sim R_A(D, A(D), D_f)$, which posses no information about $D_f$ without influencing the contributions of other data. The following definition gives the definition of existing approximate and exact unlearning methods [64, 44, 23].

**Definition 1** (Machine unlearning). *Let $A$ and $R_A$ denote the learning algorithm and the unlearning method, respectively. The pair $(A, R_A)$ achieves exact unlearning if $\forall D, D_f \subset D, A(D_r) =_d R_A(D, A(D), D_f)$, where $D_r = D \setminus D_f$ and $=_d$ means the same distribution. This means that if the unlearned model from $R_A(D, A(D), D_f)$ has no information about $D_f$, we cannot differentiate the model after forgetting from a model that is obtained on $D_r$. The pair $(A, R_A)$ satisfies $(\epsilon, \delta)$-unlearning if $\forall D, D_f \subset D,$ and $E \subset \mathbb{R}^{d_2}, P(R_A(D, A(D), D_f) \in E) \leq e^\epsilon P(A(D_r) \in E) + \delta$, where $A$ and $D_f$ denote the learning algorithm and the forget set, respectively.*

As aforementioned, the adversary aims to make malicious update requests for his/her desired attack goals. For example, the adversary could attack the targeted test samples and force them to be assigned as the attack targeted label [7]. The adversary could also have a specific target model $\theta^{tar}$ (e.g., the unfair and backdoored models) in mind and aim to induce a victim model as close as possible to that target model $\theta^{tar}$ [56]. The effective unlearning samples (i.e., $D_f \subset D$) can be obtained by solving the following formulated optimization problem

$$D_f = \arg \min_{D_f \subset D_t} \mathcal{L}_{adv}(\cdot; \theta^u(D_f)), \text{where } D_t = \{(x_p, y_p)\}_{p=1}^P \subset D. \tag{1}$$

In the above, $D_t$ represents a subset of $D$ that is accessible to the adversary, and $\theta^u(D_f)$ are the parameters found by completely eliminating specific targeted samples $D_f \subset D_t = \{(x_p, y_p)\}_{p=1}^P$. For each $x_p \in D_t$, we define a discrete indication parameter $\omega_p \in \{0, 1\}$ to indicate whether the sample $x_p$ should be completely deleted ($\omega_p = 1$) or not ($\omega_p = 0$). The forget set $D_f$ to be unlearned is denoted as $D_f = D_t \circ \Omega = \{x_p | x_p \in D_t \text{ and } \omega_p = 1\}$, where $\Omega = \{\omega_p \in \{0, 1\}\}_{p=1}^P$. Note that the above equation is a bi-level optimization problem – the minimization for $D_f$ involves the model parameters $\theta^u(D_f)$, which are themselves the minimizer of the training problem,

$$\theta^u(D_f) = R_A(D, f_D(\theta^*), D_f = D_t \circ \Omega), \text{where } D_t = \{(x_p, y_p)\}_{p=1}^P \text{and } \Omega = \{\omega_p\}_{p=1}^P. \tag{2}$$

Note that Eqn. (1) and (2) provide the high-level formulation for selecting subset $D_f$ such that the adversary's goals are maximized after unlearning. However, it is very difficult to optimize the effective update requests due to the introduced discrete indication parameters (i.e., $\Omega = \{\omega_p \in \{0, 1\}\}_{p=1}^P$) of the first constraint in Eqn. (2). Next, we take the second-order unlearning strategy proposed in [64] as *an illustrative example* to show how to solve the above formulated optimization problem. This unlearning strategy uses the inverse Hessian matrix of the second-order partial derivatives to change the original model's parameters to obtain the unlearned model [64]. For $D_t = Z = \{z_p\}_{p=1}^P \subset D$, we use $\tilde{D}_t = \tilde{Z} = \{\tilde{z}_p\}_{p=1}^P$ to denote its corresponding unlearned versions, where $\tilde{z}_p = (x_p - \xi_p, y_p)$ and $\xi_p$ is the unlearning modification for $x_p$. Following [64], we calculate the second-order change $\Delta(Z, \tilde{Z})$ by calculating all the gradients difference between $Z$ and $\tilde{Z}$ with a weighting change from the inverse Hessian of the loss function, i.e., $\Delta(Z, \tilde{Z}) = H_{\theta^*}^{-1}(\sum_{\tilde{z}_p \in \tilde{Z}} \omega_p * \nabla_\theta \ell(\tilde{z}_p; \theta^*) - \sum_{z_p \in Z} \omega_p * \nabla_\theta \ell(z_p; \theta^*))$, where $H_{\theta^*}^{-1}$ is the inverse Hessian matrix and $\omega_p \in \{0, 1\}$ denotes whether $z_p$ should be completely erased. To address the aforementioned challenge, we propose to relax each discrete variable $\omega_p$ into a continuous one of range $[0, 1]$, i.e., $\omega_p \in [0, 1]$, and then approximate the original update objective for this second-order strategy by the following one

$$\theta^u \leftarrow \theta^* - H_{\theta^*}^{-1}[\sum_{p=1}^P (\frac{1}{1 + \exp(-\varphi(2 * \omega_p - 1))}) * (\nabla_\theta \ell(\tilde{z}_p; \theta^*) - \nabla_\theta \ell(z_p; \theta^*))], \tag{3}$$

where $\ell$ is a training loss, and $\omega_p \in [0, 1]$. Here, we rewrite $\omega_p$ as $\frac{1}{2}(1 + \text{sgn}(2 * \omega_p - 1))$. Based on the fact that function $h_1(x) = \frac{1}{2}(1 - \text{sgn}(x))$ can be approximated by function $h_2(x) = 1 - \frac{1}{1 + \exp(-\varphi x)}$, we can obtain the above equation. The parameter $\varphi$ in $h_2$ represents the steepness of the curve. Additionally, the continuous property of $h_2$ allows us to solve the formulated optimization to perform selective forgetting attacks via the second-order unlearning strategy in [64]. From Section IV in [64], we can know that if $\tilde{Z} = \emptyset$, the adversary has the option to completely remove the targeted sample or retain it, based on whether $\omega_p \geq 0.5$ or $\omega_p < 0.5$. When $\tilde{Z} \neq \emptyset$, the adversary can intentionally and maliciously modify the targeted training samples via partially unlearning some data information based on Eqn. (3). Notably, following Lemma 1, we can easily generalize the above proposed attack framework to the scenario where the adversary wants to maliciously erase some features [64]. The algorithm and generalization to other unlearning methods are deferred to the supplementary material.

**Lemma 1** ([64]). *Let $F$ denote the features to be unlearned. Let $\theta^*_{-F}$ denote the optimal model retrained on the new dataset that is derived by removing the features $F$ from $D$. For learning models processing inputs $x$ using the linear transformations of the form $\theta^T x$, we have $\theta^*_{-F} \equiv \theta^*_{F=0}$, where $\theta^*_{F=0}$ is retrained by setting the values of the features $F$ to zero in $D$.*

**Definition 2** (Strongly convexity). *A function $\psi : \mathbb{R}^{d_3} \to \mathbb{R}^{d_4}$ is said to be $M$-strongly convex for some $M \geq 0$ if for any $z_1 \in \mathbb{R}^{d_3}, z_2 \in \mathbb{R}^{d_3}$, and any $q \in (0, 1)$, $\psi(q z_1 + (1 - q) z_2) \leq q\psi(z_1) + (1 - q)\psi(z_1) - \frac{M}{2} q(1 - q) ||z_1 - z_2||_2^2$. Note that if the above condition is satisfied for $M = 0$, we refer to the function $\psi$ as convex.*

**Definition 3** (Lipschitzness continuity). *A general function $\psi : \mathbb{R}^{d_3} \to \mathbb{R}^{d_4}$ is L-globally Lipschitz continuous if for all $z_1 \in \mathbb{R}^{d_3}, z_2 \in \mathbb{R}^{d_3}, ||\psi(z_1) - \psi(z_2)|| \leq L||z_1 - z_2||_2$.*

**Theorem 1.** *Let $\theta_D^* = \arg\min_{\theta \in \Theta} \ell_D(\theta)$ for any given dataset $D$. Suppose that the loss function $\ell_z$ is L-globally Lipschitz continuous and M-strongly convex for any $z \in \mathcal{Z}$. For any integer $N$, dataset $D$ of size $N$, and forget set $D_f$, we can have that $||\theta_D^* - \theta_{D \setminus D_f}^*||_2 \leq \frac{2L}{MN}|D_f|$, where $|D_f|$ represents the size of the forget set.*

The above lemma shows that we can erase features from many learning models by first setting them to zero [64]. The above theorem relates the difference of the model parameters to the forget set, and can be easily generalized to the defined data update requests in Definition 4.

## 3.2 Sequential and Dynamic Selective Forgetting Attacks

In practice, the model owner usually receives sequential update requests from one or more data owners at different times, and is asked to update the model from these sequential data update requests. Compelled by enticing incentives, the adversary could interact with this sequential update process to dynamically craft malicious update requests according to the model states, which can pose potential threats to the system. For the threat model, we here consider a very restricted setting where the adversary does not own any training data. Unlike the above static attacks, such a sequential update scenario presents a crucial challenge of transiency. Specifically, at each time step, the adversary needs to make an irrevocable decision on whether to attack, and if he/she fails, or opts not to attack, then that data point is no longer available for further attacks. In the sequential setting, the model owner receives data update requests (i.e., $\{u_t\}_{t \geq 1}$) from the users sequentially, and is asked to update the model from these update requests. We let $\mathcal{Z} = \{\mathcal{Z}, \emptyset\}$ with a slight abuse of notation. Here, the $t$-th update request $u_t$ is a tuple $u_t = (o_t, z_t^{tra}, z_t^{new})$, where $z_t^{tra} \in \mathcal{Z}$, $z_t^{new} \in \mathcal{Z}$, and $o_t \in \mathcal{O} = \{$*"Delete"*, *"Add"*, *"Modify"*$\}$ is a update instruction. Using these update requests, we can sequentially update the dataset and model as defined below.

**Definition 4** (Update sequences and sequentially updated models). *Let $\mathcal{U} = (u_1, u_2, \cdots, u_t, \cdots)$ denote the update sequence, where $u_t \in \mathcal{O} \times \mathcal{Z} \times \mathcal{Z}$ for all $t$. Given the dataset $D$ and the $t$-th update request $u_t$, the update operation for $D_t = D_{t-1} \circ u_t$ is defined as follows*

$$\begin{cases} D_t = D_{t-1} \setminus z_t^{tra}, & \text{if } (o_t = \text{"Delete"}) \wedge (z_t^{tra} \in D_{t-1}) \wedge (z_t^{new} = \emptyset) \\ D_t = D_{t-1} \cup z_t^{new}, & \text{if } (o_t = \text{"Add"}) \wedge (z_t^{tra} = \emptyset) \wedge (z_t^{new} \notin D_{t-1}) \\ D_t = (D_{t-1} \setminus z_t^{tra}) \cup z_t^{new}, & \text{if } (o_t = \text{"Modify"}) \wedge (z_t^{tra} \in D_{t-1}) \wedge (z_t^{new} \notin D_{t-1}). \end{cases} \quad (4)$$

*We write $D_0 = D$. For any $t \geq 1$, we write $\theta_t$ for the model input to the unlearning algorithm $R_A$ on time step $t$. We write $\theta_1 = A(D_0)$, and for any $t \geq 1$, $\theta_{t+1} = R_A(D_{t-1}, \theta_t, u_t)$. We write $\{D_t\}_{t \geq 0}$ to represent the sequence of updated datasets, $\{\theta_t\}_{t \geq 1}$ for the sequence of input models to $R_A$.*

The update sequences mentioned above align with the data update requests specified in [44, 23, 36], which focus on the *add* and *delete* requests. However, it is important to note that we extend these update requests to include the *modify* requests [64, 37] as well, and this extension is necessary because some existing works also incorporate the *modify* update requests to enable the deletion of partial data information [64, 37, 36, 2, 41, 17]. The adversary's goal is to force current updated model to satisfy certain desired properties at each time step while paying a small cost. For example, the adversary wants to force the current updated model to approach or maintain a target model $\theta^{tar}$. Then we can define the adversary's goal as $\mathcal{L}_{adv} = ||\theta - \theta^{tar}||$. We propose to formulate the sequential selective forgetting attacks as a Markov Decision Process (MDP) $\mathcal{M} = (\mathcal{S}, \mathcal{A}, \mathcal{T}, \mathcal{R}, \gamma)$, where

- $\mathcal{S}$ is the state space. The state $s_t$ at time step $t$ is the stacked vector $s_t = [\theta_t, u_t]^T$ consisting of the current model $\theta_t$ and the incoming update request $u_t$, where $u_t \in \mathcal{O} \times \mathcal{Z} \times \mathcal{Z}$. The state space is $\mathcal{S} = \Theta \times \mathcal{O} \times \mathcal{Z} \times \mathcal{Z}$. We assume that the initial model $\theta_0$ is fixed and known to the adversary while the first update request $u_1$ is sampled from $P$, i.e., the initial state distribution is defined as $\mu_0(\theta_0, u_1) = P(u_1)$.

- $\mathcal{A}$ is the adversary action space. For each update request $u_t$, we define a discrete indication parameter $k_t \in \{0, 1\}$ to denote whether at time step $t$ the attack action should applied ($k_t = 1$) or not ($k_t = 0$). If the update instruction $o_t$ for $u_t$ is "*Add*", we assume that the adversary can only introduce imperceptible perturbations $a_t$ with the purpose of manipulating $z_t^{new}$ for stealthiness. Thus, the attacked $u_t$ can be represented as $\tilde{u}_t = (o_t, z_t^{tra} = \emptyset, \tilde{z}_t^{new} =$

$z_t^{new} + k_t \cdot a_t$). When $o_t =$"*Modify*" and $k_t = 1$, we can write the attacked $u_t$ as $\tilde{u}_t = (o_t, z_t^{tra}, \tilde{z}_t^{new} = z_t^{new} + k_t \cdot a_t)$, where $a_t$ is the imperceptible perturbations. When $u_t$ is a *delete* update request and the adversary chooses to attack this *delete* update request (i.e., $k_t = 1$), we assume that the adversary has the ability to manipulate the update instruction by converting $o_t =$"*Delete*" into $\tilde{o}_t =$"*Modify*", and the attacked $u_t$ is $\tilde{u}_t = (\tilde{o}_t, z_t^{tra}, \tilde{z}_t^{new} = z_t^{tra} + k_t \cdot a_t)$. Here, the adversary maliciously modifies $z_t^{tra} \in D_{t-1}$ (instead of directly deleting $z_t^{tra}$). Note that the adversary can also choose not to attack this *delete* request.

- $\mathcal{T}$ denotes the state transition function. The state transition function $\mathcal{T} : \mathcal{S} \times \mathcal{A} \to \Delta_{\mathcal{S}}$ represents the conditional probability of the next state given the current state and attack action. We assume that the update function $g$ is deterministic. Therefore, the stochasticity arises solely from $u_{t+1}$ within $s_{t+1}$. To provide a concrete example, we consider the modification step where $o_t$ corresponds to the action "*Modify*". In this case, the transition function can be derived as

$$
\mathcal{T}(s_{t+1}|s_t, a_t) = \mathcal{T}(u_{t+1}, \theta_{t+1}|u_t, \theta_t, a_t) = P(u_{t+1}|u_t, \theta_t, a_t)Pr(\theta_{t+1} = g(u_t, \theta_t, a_t))
$$
$$
= P(u_{t+1}|u_t, \theta_t, a_t)Pr(\theta_{t+1} = g'(\tilde{u}_t, \theta_t)) = P(u_{t+1}|u_t, \theta_t, a_t)Pr(\theta_{t+1} = R_A(D_{t-1},
$$
$$
\theta_t, \tilde{u}_t)) = P(u_{t+1}|u_t, \theta_t, a_t) = P(u_{t+1}). \tag{5}
$$

Note that in the above *modify* case, given the attack action $a_t$, we can obtain $\tilde{u}_t = (o_t, z_t^{tra}, \tilde{z}_t^{new} = (z_t^{new} + a_t))$. Discussions on other data update cases (also the proposed sequential optimization approach) can be found in the supplementary material.

- $\mathcal{R}$ is the cost function. We define the cost at the time step $t$ as $\mathcal{R}(s_t = [\theta_t, u_t]^T, k_t, a_t) := \|\theta_{t+1} - \theta^{tar}\|_2$, which is determined by the current state and the attack action.

A policy is a function $\Phi_{\mathcal{M}} : \mathcal{S} \to \mathcal{A}$ that the adversary uses to choose the attack action $a_t = \Phi_{\mathcal{M}}(s_t = [\theta_t, u_t]^T)$ based on the current victim model $\theta_t$ and update request $u_t$. Note that $k_t \in \{0, 1\}$ denotes whether at time step $t$ the attack action should be applied. Now, the problem is how the adversary can find an effective attack strategy $\{k_t\}_{t\geq 1}$ with the corresponding attack action $a_t$ (generated by the policy network), which can maximize the adversary's goal (i.e., $\mathbb{E}_{\mathcal{M}} \sum_{t=0}^{\infty} -\gamma^t \mathcal{R}(s_t, k_t, \Phi_{\mathcal{M}}(s_t)))$ and minimize the number of attacked time steps (i.e., $\sum_{t=1}^{T} k_t$) for stealthiness. A naive way is to attack each step. However, attacking all the time steps would cause suspicion and expose the identity of the adversary. To address this, we formulate the below optimization to obtain the effective policy

$$
\min_{\Phi_{\mathcal{M}}, \{k_t\}_{t\geq 1}} \mathbb{E}_{s\sim\mu_0}\mathbb{E}_{\mathcal{M}} \sum_{t=0}^{\infty} \gamma^t \mathcal{R}(s_t = [\theta_t, u_t]^T, k_t, a_t = \Phi_{\mathcal{M}}(s_t)) + \sum_{t=1}^{\infty} k_t
$$
$$
\text{s.t., } \theta_t = R_A(D_{t-2}, \theta_{t-1}, u_{t-1}), k_t \in \{0, 1\}, \tag{6}
$$

where $\Phi_{\mathcal{M}}$ is the policy network to be optimized. In the above first constraint, we use the notation $u_t$ without explicitly distinguishing whether the $t$-th update request is subjected to an attack or not.

However, directly solving the above optimization problem is highly challenging due to the involvement of numerous variables. In addition, the environment data distribution for the formulated MDP is fixed but unknown to the adversary. To address the first challenge, we propose training an adversarial policy network $\Phi^{adv}$ that takes the current state $s_t$ as input and outputs the attack strategy $(p_t, a_t')$, where $p_t$ is the probability for taking the malicious action $a_t'$. Specifically, for each step, the adversary gets the attack strategy $(p_t, a_t')$ from the adversarial policy (i.e., $\Phi^{adv} : s_t \to (p_t, a_t')$). If $p_t \geq 0.5$, the adversary designates step $t$ as the critical point and introduces perturbations to mislead the model owner to trigger the action $a_t'$. Otherwise, the adversary does not attack the current time step. Additionally, the adversary can construct a progressively refined empirical distribution $\hat{P}_t$ based on the sequence of observations $u_{1:t}$. More precisely, at time $t$, by replacing $P$ with $\hat{P}_t$ and the model $\theta_0$ with $\theta_t$, the attacker can construct a substitute MDP $\hat{\mathcal{M}}_t = (\mathcal{S}, \mathcal{A}, \hat{\mathcal{T}}_t, \mathcal{R}, \gamma)$, solve for the optimal policy $\Phi^*_{\hat{\mathcal{M}}_t}$ on $\hat{\mathcal{M}}_t$, and then apply the learned policy to perform the one-step attack. In this paper, we solve the substitute MDP using deep deterministic policy gradient (DDPG) [33, 35] to handle a continuous action space.

**Theorem 2.** *Let $\Phi^*_{\mathcal{M}}$ and $\Phi^*_{\hat{\mathcal{M}}}$ be the optimal policies for $\mathcal{M}$ and $\hat{\mathcal{M}}$ respectively, with the same initial state distribution $\mu_0$. We assume that the 1-Wasserstein distance between the estimated distributions $\hat{P}$ and the true distribution $P$ satisfies $W_1\left(\hat{P}, P\right) \leq \upsilon$. Additionally, we assume the*

*loss function $\ell : S \times \mathcal{Z} \to \mathbb{R}$ exhibits Lipschitz continuity with respect to both s and z, with a constant $L_1$, and Lipschitz smoothness with respect to z, with a constant $L_2$. Moreover, we assume the loss function to possess strong convexity, strong smoothness, and twice continuous differentiability with respect to s. Let $\mathcal{J}_{\mathcal{M}}(\Phi_{\mathcal{M}})$ be the cost in Eqn. (6). Then there exist two constants $\Psi$ and $\Omega$ such that:*

$$\left| \mathcal{J}_{\mathcal{M}}\left(\Phi_{\mathcal{M}}^*\right) - \mathcal{J}_{\mathcal{M}}\left(\Phi_{\hat{\mathcal{M}}}^*\right) \right| \leq \upsilon\Omega(L_1(L_2\zeta + 2) + \Psi L_2\zeta), \tag{7}$$

*where $\zeta$ is a constant related with updating the model.*

**Discussions.** The above theorem implies that the difference between $\mathcal{J}_{\mathcal{M}}\left(\Phi_{\mathcal{M}}^*\right)$ and $\mathcal{J}_{\mathcal{M}}\left(\Phi_{\hat{\mathcal{M}}}^*\right)$ depends on the distribution difference $\upsilon$. In the above, we discuss our proposed selective forgetting attacks in the white-box setting. In the black-box setting, the adversary can randomly select substitute models and exploit the transferability property [11, 45, 58], which arises from the shared vulnerabilities or decision boundaries among different models. For instance, in approximate unlearning methods, the adversary can train one or several models to substitute the well-trained model $\theta^*$. This allows the adversary to generate malicious update requests and effectively transfer them to the target black-box victim model.

## 4 Experiments

In this section, we conduct comprehensive experiments to evaluate the effectiveness of our proposed selective forgetting attacks. All the experiments are run for 10 individual trials with different random seeds. Due to space limitations, a detailed description of the experimental setup, parameter settings, and more experimental results are given in the supplementary material.

**Datasets and models.** In experiments, we adopt the following real-world datasets: CIFAR-10 [31], Adult [15], Diabetes [1], and MNIST [12]. CIFAR-10 consists of 60,000 color images across 10 classes. Adult is downsampled to 23,374 samples with 57 features. Diabetes comprises 70,692 survey responses with 21 numerical features. MNIST contains 70,000 grayscale images of handwritten digits. In addition, we adopt a synthetic dataset, which is a binary dataset with 10,000 samples and 20 features and is generated based on the normal distribution. We also use a range of machine learning models, including the logistic regression model, ResNet-18 [25], VGG-16 [53], MobileNetV2 [50], and a neural network with two fully connected layers.

**Attack settings.** In experiments, we implement malicious selective forgetting attacks using the following unlearning methods: first-order based [64], second-order based [64], unrolling SGD [57], amnesiac [21], and SISA [4]. In static forgetting attacks, we first pre-train the model and then perform selective forgetting attacks in targeted and untargeted settings. In the targeted setting, the adversary aims to misclassify the input as a specific target class, while in the untargeted setting, the adversary aims to mislead the model into predicting any incorrect class. In sequential forgetting attacks, we first generate the target model from static forgetting attacks and then force the victim model to be close to the target model. In experiments, we allow DDPG [35] to train once at the beginning to learn the optimal policy on the pre-attack data and then apply the learned policy to perform the one-step attack.

**Baselines.** In experiments, we adopt the *RandSearch* baseline, where we randomly select a set of training samples to be forgotten in static forgetting attacks. In sequential selective forgetting attacks, we consider the *no attack* and *random attack* baselines. Specifically, the *no attack* baseline keeps the incoming data update requests unchanged, and the *random attack* baseline adds random data noise to modify the update requests.

### 4.1 Experimental Results for Static Selective Forgetting Attacks

First, we conduct experiments to investigate the performance of malicious selective forgetting attacks in a static manner. We adopt the attack success rate as the evaluation metric, defined as the number of successful attacks achieved among all attack attempts. For $\mathcal{L}_{adv}$ in Eqn. (1), we adopt the $f_6$ function in [7]. Table 1 summarizes the attack success rate of static forgetting attacks in the targeted setting via first-order, second-order, unrolling SGD, amnesiac, and SISA. Our proposed methods consistently achieve high attack success rates across diverse datasets and unlearning procedures. For example, our proposed methods hit an attack success rate of 1.0 on Diabetes and 0.8 on CIFAR-10. The reason is that our proposed methods can assign importance scores to training samples based on

Table 1: Attack success rate of static forgetting attacks in the targeted setting.

| Unlearning method | Diabetes | | CIFAR-10 | |
| --- | --- | --- | --- | --- |
| | RandSearch | **Ours** | RandSearch | **Ours** |
| First-order | $0.04 \pm 0.03$ | $\mathbf{1.00 \pm 0.00}$ | $0.08 \pm 0.04$ | $\mathbf{0.80 \pm 0.04}$ |
| Second-order | $0.06 \pm 0.04$ | $\mathbf{1.00 \pm 0.00}$ | $0.10 \pm 0.08$ | $\mathbf{0.82 \pm 0.04}$ |
| Unrolling SGD | $0.04 \pm 0.03$ | $\mathbf{1.00 \pm 0.00}$ | $0.06 \pm 0.03$ | $\mathbf{0.78 \pm 0.06}$ |
| Amnesiac | $0.08 \pm 0.04$ | $\mathbf{0.98 \pm 0.02}$ | $0.04 \pm 0.03$ | $\mathbf{0.74 \pm 0.08}$ |
| SISA | $0.47 \pm 0.12$ | $\mathbf{0.74 \pm 0.07}$ | $0.13 \pm 0.05$ | $\mathbf{0.67 \pm 0.08}$ |

the impact of the targeted loss and unlearn optimal samples, resulting in substantial enhancements in attack performance. In contrast, the RandSearch baseline performs poorly in identifying the training samples to be removed for misclassifying the targeted test samples. These experimental results show the applicability and effectiveness of our optimization framework for static selective forgetting attacks across various unlearning methods, enabling us to achieve desired attack goals.

Next, we study the impact of underrepresentation, which occurs when certain classes in a dataset have fewer instances than others. In Figure 2a, we unlearn $1\%$ to $5\%$ of training samples from CIFAR-10. In Figure 2b, we unlearn $5\%$ to $25\%$ of training samples from Diabetes. The unlearning samples are exclusively selected from the same class as the target class. We compare the attack performance with and without our proposed optimization framework. As illustrated, the attack success rates increase correspondingly with the percentage of

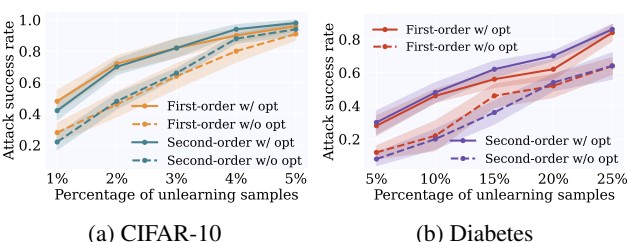

(a) CIFAR-10      (b) Diabetes

Figure 2: Attack success rate of selective forgetting attacks when data is underrepresented, comparing with and without optimization. The shaded area represents the standard error.

unlearning samples, as removing certain training data can hinder the model's ability to learn the target class. However, randomly removing a small portion of data without optimization has a minimal effect on the attack success rates. In contrast, our proposed methods can effectively identify the most critical training samples to be removed, even within a specific class, enhancing the attack performance. Therefore, malicious selective forgetting attacks can leverage data underrepresentation to achieve the desired attack goals.

Then, we examine the impact of subpopulations, which are groups of samples having similar features. Here, we focus on misclassifying a particular subpopulation of targeted test samples. Upon identifying a test sample for a successful attack, we create a cluster consisting of points most similar to that sample based on the final representation layer in the network. We then apply the same malicious update requests to attack subpopulations of varying sizes. As a baseline comparison, we randomly select group mem-

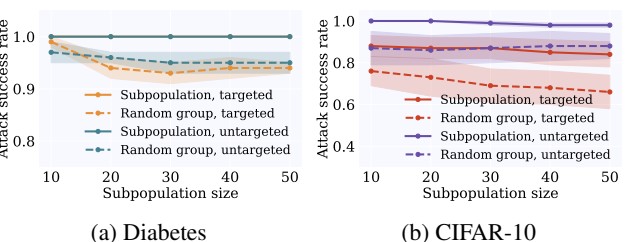

(a) Diabetes      (b) CIFAR-10

Figure 3: Attack success rate of selective forgetting attacks across subpopulations of varying sizes.

bers of the same sizes. In Figure 3, we adopt unrolling SGD on Diabetes and amnesiac on CIFAR-10. We observe that malicious selective forgetting attacks are highly effective in attacking subpopulations of test samples in both targeted and untargeted settings. For instance, when using the unrolling SGD to attack a subpopulation of size 50 on Diabetes, the targeted attack success rate remains at 1.0. Similarly, with the amnesiac on CIFAR-10, the targeted attack success rate only drops to 0.85 for a subpopulation size of 50. It is worth noting that while the baseline randomly generated groups are not as good as those formed based on similarity, the attack performance is still impressive.

Further, we explore the impact of data diversity on malicious selective forgetting attacks. We apply PCA projection on input features and cluster the training data for each class. Then we formulate diversity by sampling different percentages of data points for each cluster. In Figure 4, we unlearn the same number of data points in each cluster and evaluate the attack performance with random selection and our proposed optimization framework in targeted and untargeted settings. Firstly, our proposed methods demonstrate the ability to identify the most influential data points within each cluster, leading to substantial improvements in attack success rates compared to the random removal of data points. Secondly, removing an equal number of critical data points

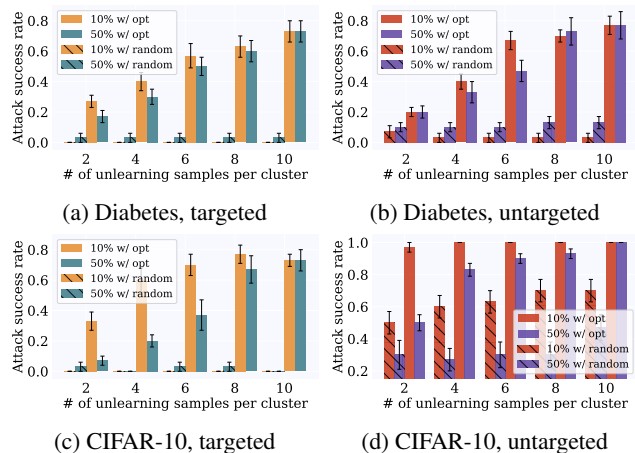

(a) Diabetes, targeted     (b) Diabetes, untargeted

(c) CIFAR-10, targeted     (d) CIFAR-10, untargeted

Figure 4: Attack success rate of selective forgetting attacks with clusters of $10\%$ data points (less redundant) and with clusters of $50\%$ data points (more redundant).

from less redundant clusters has a greater impact on the unlearned model, resulting in higher attack success rates than more redundant clusters. Consequently, unlearning training data in different scopes of diversity can affect the performance of malicious selective forgetting attacks.

## 4.2 Experimental Results for Sequential Selective Forgetting Attacks

In this section, we evaluate the performance of malicious selective forgetting attacks in the sequential update setting. We adopt the Euclidean distance metric to quantify the similarity between the victim model and the target model. Firstly, we examine the model convergence over attack time steps, where we naively apply attacks at each time step in a sequence of update requests. In Figure 5, we incorporate *add*, *delete*, and *modify* requests in a sequence of 300 update requests using the first-order unlearning method. As shown in the figure, our attack steadily reduces the Euclidean distance between the victim model and the target model over time steps on each adopted dataset (we use digits 1 and 7 in MNIST). In contrast, the random attack baseline and the no attack baseline fail to approach the target model. Our learned adversarial policy demonstrates the ability to attack the sequential data update requests by selecting the most effective actions within a small difference to the requests, resulting in a minor difference between the updated model and the target model.

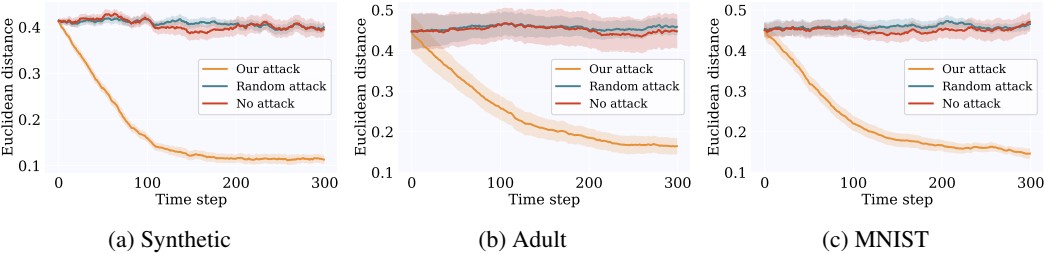

(a) Synthetic     (b) Adult     (c) MNIST

Figure 5: Euclidean distance of the victim model to the target model at each time step.

Next, we investigate the performance variations corresponding to different attack steps within the sequential update process. In Figure 6, we compare corresponding models under different numbers of attack times with the optimal model (which is obtained by attacking all time steps and converges to the target model as shown in Figure 5). The results show that our attack method consistently diminishes the Euclidean distance between the victim model and the optimal model as the number of attack times increase. Remarkably, our optimally leaned policy injects perturbations on some specific update requests only when necessary, leading to fewer attack times and achieving comparable performance as the optimal model. However, the random attack baseline does not contribute much to approaching the optimal model as the number of attack times increase. Therefore, our proposed

optimization framework proves stealthiness and effectiveness in identifying the critical attack time steps and inducing the effective attack actions to the sequential update requests.

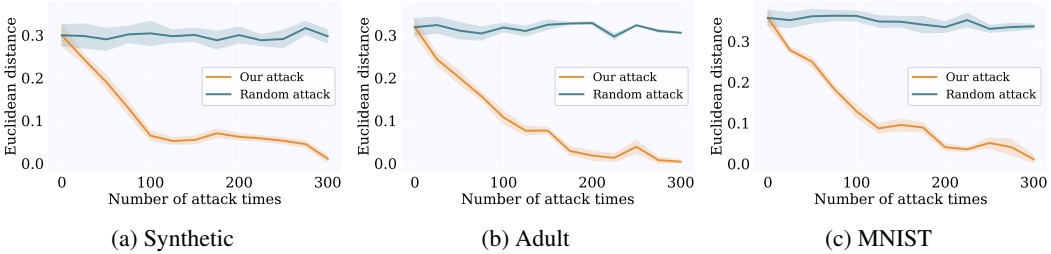

(a) Synthetic          (b) Adult          (c) MNIST

Figure 6: Euclidean distance of the victim model to the target under different numbers of attack times.

## 4.3 Black-box Experiments

In this section, we consider malicious selective forgetting attacks in the black-box setting. Firstly, we investigate the transferability of selective forgetting attacks across different machine learning models. In Table 2, we employ ResNet-18, VGG-16, and MobileNetV2 in the untargeted setting. As illustrated, our proposed methods demonstrate the ability to transfer the generated malicious update requests to attack the black-box model, even though the black-box model is trained with a different model than the substitute model. For example, ResNet-18 can achieve attack success rates of 0.86 and 0.75 when transferred to VGG-16 and MobileNetV2, respectively. Furthermore,

Table 2: Attack success rates in the black-box setting using substitute models on CIFAR-10.

| Black-box 
 Substitute | ResNet-18 | VGG-16 | MobileNetV2 |
|---|---|---|---|
| ResNet-18 | $1.00 \pm 0.00$ | $0.86 \pm 0.05$ | $0.75 \pm 0.07$ |
| VGG-16 | $1.00 \pm 0.00$ | $1.00 \pm 0.00$ | $0.68 \pm 0.05$ |
| MobileNetV2 | $0.98 \pm 0.02$ | $0.72 \pm 0.09$ | $1.00 \pm 0.00$ |

Table 3: Attack success rate in the black-box setting using substitute unlearning methods on Diabetes.

| Black-box 
 Substitute | First-order | Second-order | Unrolling SGD |
|---|---|---|---|
| First-order | $1.00 \pm 0.00$ | $0.88 \pm 0.08$ | $0.96 \pm 0.03$ |
| Second-order | $1.00 \pm 0.00$ | $1.00 \pm 0.00$ | $0.94 \pm 0.06$ |
| Unrolling SGD | $1.00 \pm 0.00$ | $0.94 \pm 0.03$ | $1.00 \pm 0.00$ |

we investigate the transferability of selective forgetting attacks between different unlearning methods. In Table 3, we apply first-order, second-order, and unrolling SGD unlearning methods in the targeted setting. We observe that malicious update requests generated by our proposed methods can also be effectively transferred to different unlearning methods in the black-box setting. The reason is that with the same update requests, the unlearned models share similar decision boundaries among different models and unlearning methods.

## 5 Conclusion

In this paper, we examine the security vulnerability and resilience of machine learning models against selective forgetting attacks during the unlearning process, without compromising the integrity of the training and testing procedures, as commonly seen in traditional evasion attacks and data poisoning attacks. Specifically, we first present the general framework for selective forgetting attacks in the static setting, enabling the adversary to generate malicious whole data deletion requests. Additionally, we propose a new approach for developing sequential selective forgetting attacks that can effectively compromise the unlearning system using the sequential data update requests. We also conduct theoretical analysis for the proposed selective forgetting attacks. The reported extensive experimental results demonstrate the effectiveness of our proposed selective forgetting attacks. We believe that our findings provide valuable insights into how to design secure and robust mechanisms to defend against selective forgetting attacks in the future.

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
