# OpenReview forum: "Static and Sequential Malicious Attacks in the Context of Selective Forgetting"
_NeurIPS.cc/2023/Conference — NeurIPS 2023 poster_

### Official Review · Reviewer_MTyE · 2023-06-14

**Soundness:** 3 good
**Presentation:** 2 fair
**Contribution:** 3 good
**Rating:** 4
**Confidence:** 4

**Summary:**

This paper investigates the potential and viability of malicious data update requests in the context of the unlearning process. The authors put forward a malicious selective forgetting attack in a static scenario and present a framework for sequential forgetting attacks. And the framework is formulated as a stochastic optimal control problem.

**Strengths:**

This paper investigates the potential and viability of malicious data update requests in the context of the unlearning process.


**Weaknesses:**

The attack goal of the proposed approach remains unclear.

The attack methodology of the proposed approach lacks clarity.

The authors assert the effectiveness of their method in both white-box and black-box scenarios. However, the methodology section lacks a clear explanation of how the attack steps differ in these two scenarios.

More experiments on high-resolution datasets, such as ImageNet, VGG-Flower, that include more classes, should be evaluated. Otherwise, it is not sure whether the proposed method is applicable in the real-world scenario.

**Questions:**

The attack goal of the proposed approach remains unclear. While the paper discusses constructing malicious data update requests during the unlearning process, it does not explicitly define the specific objectives pursued by the attacker. It is essential for the authors to provide concrete explanations regarding the attack goals, rather than merely stating them as desired outcomes. For instance, is the intention to diminish the fairness of the unlearning model or to deliberately misclassify specific samples?Furthermore, the paper lacks a high-level description of how the attack is executed, given a clear attack target. It is important for the authors to outline the general methodology employed by the attacker to carry out the attacks.

The attack methodology of the proposed approach lacks clarity. While the authors present multiple definitions, lemmas, and theorems in the methodology section, they do not provide an overall step-by-step explanation of how the attack is conducted. As a result, obtaining a clear understanding of the attack process remains challenging.

An overview image is also required to describe the proposed attacks.

The authors assert the effectiveness of their method in both white-box and black-box scenarios. However, the methodology section lacks a clear explanation of how the attack steps differ in these two scenarios.

More experiments on high-resolution datasets, such as ImageNet, VGG-Flower, that include more classes, should be evaluated. Otherwise, it is not sure whether the proposed method is applicable in the real-world scenario.



**Limitations:**

Yes

---

> ### Author Rebuttal · Authors · 2023-08-09
>
> We really appreciate your thoughtful comments and valuable suggestions. Below, we provide our response to the questions and concerns.
>
> **1: Clarifying the attack goal (e.g., diminishing the fairness of the unlearning model, or misclassifying specific samples?) of the proposed approach. Giving a high-level description of how the attack is executed.**
>
> First, we want to clarify that our proposed attacks are general and can offer great flexibility to accommodate the adversary's desired diverse attack objectives through the malicious unlearning samples (Please refer to lines 136-138 in Section 3.1 of the main manuscript). In this paper, we present novel static and sequential selective forgetting attack frameworks that empower the adversary to achieve desired attack goals, such as causing unfairness for specific groups of individuals or inducing misclassifications, by crafting malicious data update requests.
>
> Additionally, the high-level descriptions of the proposed attacks are deferred to Section 4 in the Supplementary Material due to space limitations during the preparation of our submission. For example, in Algorithm 1 presented in the Supplementary Material, we illustrate the high-level descriptions of how to solve the proposed attack optimization framework using the second-order unlearning strategy described in Eqn. (1) and (3) of the main manuscript. In the final version, we will integrate these high-level descriptions for the attacks from the Supplementary Material into the main manuscript.
>
> **2: Clarifying the attack methodology of the proposed approach by providing the step-by-step explanations of how the attack is conducted.**
>
> Due to space limitations during the preparation of our submission, in Section 4 of the Supplementary Material, we provide the algorithms (e.g., Algorithm 1, 2, 3, and 4) to give detailed explanations of how the attacks are conducted in the static and sequential settings. In the final version, we will incorporate these high-level descriptions from the Supplementary Material into the main manuscript.
>
> Additionally, in the one-page PDF in the "global" response, we have included the attack flowchart (please refer to Figure 2 and Figure 4 in this one-page PDF) for our attacks in the static and sequential settings.
>
> **3: Providing an overview image to describe the proposed attacks.**
>
> Thanks for pointing this out. In the attached one-page PDF in the “global” response, following your suggestion, we provide an overview image (please refer to Figure 1 and Figure 3  in this one-page PDF) to describe the proposed attacks in the static and sequential attack settings.
>
> **4: Giving more explanations of how the attack steps differ in the black-box and white-box scenarios.**
>
> Thanks for this suggestion. In our paper, we implement our selective forgetting attacks in both white-box and black-box scenarios. In the white-box setting, we assume that both the model architecture and parameters are known to us. Leveraging this knowledge, we can directly generate malicious data update requests for specific test samples on the pre-trained model. In the black-box attack, as we don't know any information about the model, we first need to train one or several surrogate models to substitute the pre-trained model, and then transfer selective forgetting attacks to the black-box victim model. In detail, we generate malicious data update requests on one model and apply these malicious data update requests to attack the black-box victim model.
>
> **5: Including more experiments on high-resolution datasets, such as ImageNet, and VGG-Flower.**
>
> We greatly appreciate the suggestions for adding experiments on ImageNet and VGG-Flower. Following your suggestions, we conducted experiments on ImageNet [r1] and Flowers [r2], with high-resolution images and more classification classes. In experiments, we trained ResNet-18 on a sub-dataset of ImageNet and VGG-19 on Flowers. Following the same experimental setting in Section 4.1 of the main manuscript, we report the attack success rate of our proposed static forgetting attacks in the untargeted setting, and we compare the results with the RandSearch baseline. As shown in the below table, our proposed methods also achieve high attack success rates on these two datasets using the first-order and the rolling SGD unlearning methods. On the other hand, the RandSearch baseline demonstrates limited effort in misclassifying targeted test samples in an untargeted manner. In the final version, we will include the experiments on high-resolution datasets.
>
> Table 1: Attack success rate of our proposed static forgetting attacks on ImageNet and Flowers.
> | Dataset  | Unlearning method | RandSearch | Ours |
> | :---: | :---: | :---: | :---: |
> | ImageNet | First-order | 0.26 ± 0.08 | 0.96 ± 0.04 |
> | ImageNet | Unrolling SGD | 0.24 ± 0.07 | 0.94 ± 0.04 |
> | Flowers | First-order | 0.46 ± 0.13 | 1.00 ± 0.00 |
> | Flowers | Unrolling SGD | 0.35 ± 0.10 | 0.95 ± 0.05 |
>
> **Reference.**
>
> [r1] "Imagenet: A large-scale hierarchical image database", CVPR 2009.
>
> [r2] "Automated flower classification over a large number of classes", ICVGIP 2008.

---

### Official Review · Reviewer_EC3c · 2023-06-30

**Soundness:** 3 good
**Presentation:** 3 good
**Contribution:** 3 good
**Rating:** 6
**Confidence:** 4

**Summary:**

This paper studies the malicious data update in machine unlearning. The authors consider two strategies. The first is static selective forgetting attack framework, where the adversary exploits vulnerabilities in the unlearning systems by submitting a set of carefully crafted data update requests at once. The second is sequential selective forgetting attack framework that injects malicious update multiple times by considering the order and timing of data update requests. Theoretical and experimental analysis are provided to validate the proposed machine unlearning attacks.

**Strengths:**

•	The angle of the problem is novel as there is no study on malicious machine unlearning so far.
•	The authors propose one-step and multi-step attacks, and study the attack effect in white/black box, targeted/untargeted settings.
•	The two proposed methods are well-justified both theoretically and experimentally.


**Weaknesses:**

•	Experiment detail missing:
      o	What is target class in the targeted setting?
      o	What is the class that the machine unlearning aims to forget? The entire class or a few samples? If latter(according to line 344-346), what does forgetting mean in this case?
      o	What does the attack success rate mean, especially in the targeted setting? Reaching the targeted class or no longer on the previous class?
      o	For untargeted setting, will the random class be the class for the task of machine unlearning? If so, what is the impact?
      o	What is the setting (untargeted/targeted) for black-box attack? The authors should specify in the manuscript.
•	Related work to model poisoning should be added. While the authors mentioned data poisoning attacks, the proposed attack in machine unlearning is closer to model poisoning, which embeds the attack goal with the benign training objectives. Likewise, the proposed attack embeds the attack goal into the unlearning objective.


**Questions:**

See weakness.

**Limitations:**

Not applicable.

---

> ### Author Rebuttal · Authors · 2023-08-09
>
> We really appreciate your thoughtful comments and valuable suggestions. Below, we provide our response to the questions and concerns.
>
> **1: Providing more experiment details: (1) What is the target class in the targeted setting? (2) What is the class that the machine unlearning aims to forget? The entire class or a few samples? If the latter (according to lines 344-346), what does forgetting mean in this case? (3) What does the attack success rate mean, especially in the targeted setting? Reaching the targeted class or no longer on the previous class? (4) For an untargeted setting, will the random class be the class for the task of machine unlearning? If so, what is the impact? (5) What is the setting (untargeted/targeted) for black-box attack? The authors should specify in the manuscript.**
>
> Sorry about the unclear descriptions regarding these mentioned experiment details. Below, we address your concerns one by one.
>
> (1) In the targeted setting, the "target class" refers to the class that the adversary aims to attack. For example, the motivated adversary could generate malicious data update requests to attack the targeted test samples and force the targeted test sample (e.g., the bird image) to be assigned as the attack targeted label (e.g., the dog label).
>
> (2) To be clarified, our goal is not to forget certain classes. Instead, we aim to use machine unlearning techniques to forget specific training samples and their corresponding influence on the pre-trained model. In lines 344-346 of the main manuscript, we conducted selective forgetting attacks to misclassify the targeted test samples by unlearning a subset of training data (the percentages indicate the total portion of training samples to be unlearned).
>
> (3) The attack success rate means the number of successful attacks achieved over all attack attempts. Specifically, in the targeted setting, a successful attack means the targeted test sample is misclassified into the adversary’s specified label on the victim model.
>
> (4) In the untargeted setting, the predicted label of the targeted test sample is changed to any label other than the true label, after unlearning a subset of training data. This can lead to misbehaviors on the victim model, such as degrading the classification accuracy of specific users. Consider an example where existing unlearning techniques are utilized to repair a face recognition system. During this process, in the untargeted setting, the adversary could make malicious data update requests to cause the repaired face recognition system to misidentify their intended target as anyone else other than the true identity. Lastly, it is important to emphasize that our task is not to select a random class and then perform machine unlearning to completely forget that specific class.
>
> (5) In the black-box attack, we consider a setting where we have no prior knowledge about the target pre-trained victim model. Therefore, we explore the transferability of selective forgetting attacks across various machine learning models. In detail, we can generate malicious data update requests in the targeted setting (where the predicted label is changed to a specified one) and the untargeted setting (where the label is changed to an incorrect one) on one surrogate model, and then transfer these malicious data update requests to attack the black-box victim model.
>
> **2: Adding the related work to model poisoning attacks.**
>
> Thank you for your valuable suggestion. In the final version, we will incorporate related work on model poisoning attacks. Here, we want to clarify that our proposed selective forgetting attacks and model poisoning attacks differ significantly in terms of attack timing and attack mechanisms. Specifically, our attacks occur during the unlearning process, while model poisoning attacks usually take place during learning (e.g., [r1] assuming that the adversary directly attacks the model parameters during the learning process) or fine-tuning (e.g., [r2] involving a model fine-tuning step to attack the pre-trained model via the maliciously designed penalty term for fine-tuning). Additionally, our attacks utilize existing unlearning techniques to delete certain unlearning samples, whereas model poisoning attacks modify model parameters through means such as parameter manipulation [r1]. We appreciate this constructive suggestion and will ensure this suggested related work discussion in the final version.
>
> **Reference**
>
> [r1] “Local model poisoning attacks to Byzantine-Robust federated learning”, USENIX Security 2020.
>
> [r2] “Fooling Neural Network Interpretations via Adversarial Model Manipulation”, NeurIPS 2019.

---

### Official Review · Reviewer_MEi8 · 2023-07-05

**Soundness:** 2 fair
**Presentation:** 3 good
**Contribution:** 3 good
**Rating:** 5
**Confidence:** 5

**Summary:**

This paper explores the malicious forgetting issue in model unlearning and proposes two attack strategies: static attack and dynamic sequential attack. The authors also present a theoretical framework for selective forgetting attacks. Experimental results on multiple benchmark datasets demonstrate that the proposed attack method poses a significant security threat to model unlearning.


**Strengths:**

- Model unlearning as an effective strategy for data forgetting has garnered significant attention, making the exploration of potential malicious attacks during the unlearning process an interesting research direction.
- The paper has a clear research motivation, well-structured writing, and provides ample theoretical support.


**Weaknesses:**

- It is unclear whether this topic has been previously studied. The core of the proposed unlearning attack lies in a reasonable data sampling strategy for the victim model, which may not be novel from a technical standpoint.
- It would be interesting to investigate if the defender, being aware of the malicious unlearning procedure, how could detect or filter out such requests to counter the attack. For example, could perturbation noise be added to mitigate the malicious impact of unlearning?
- Recent study like Anti-backdoor Learning [1] uses two-stage unlearning techniques against backdoor attacks, whether this method could be applied to mitigate the negative effect of proposed unlearning attack?
- The authors should discuss the potential limitations of this attack.

I would be happy to improve the score if the authors address the aforementioned issues.

[1] Yige Li, Xixiang Lyu, Xingjun Ma, et al, Anti-Backdoor Learning: Training Clean Models on Poisoned Data, NeurIPS, 2021


**Questions:**

Refer to the weaknesses mentioned above.

**Limitations:**

The method does not exhibit apparent limitations. One main concern lies in the lack of convincing baseline comparisons in the experiments.

---

> ### Author Rebuttal · Authors · 2023-08-09
>
> We really appreciate your thoughtful comments and valuable suggestions. Below, we provide our response to the questions and concerns.
>
> **1: Discussions on whether this topic has been previously studied, and the technical standpoint.**
>
> Thank you for your helpful suggestions regarding our work's novelty. As far as we know, we are the first to investigate how the adversary can exploit selective forgetting mechanisms to wholly and sequentially delete unlearning samples in an attempt to undermine the integrity and performance of the victim model. In the Related Work Section of the main manuscript, we give detailed comparisons with existing related works.
>
> From the technical standpoint, different from traditional empirical sampling methods, the proposed static and sequential attack frameworks use discrete indication variables to formulate the complete deletion of targeted training samples, which are very hard to be directly solved due to the presence of non-differentiable loss functions. To address the challenges, we first design a continuous and differentiable function to approximate the discrete component in the formulated static attack framework, and then optimize the proposed sequential unlearning attack framework by training an adversarial policy network, which is specifically designed to target and attack a few critical data update requests in the sequential data update setting. We believe that these technical advancements strengthen the contributions made in this paper.
>
> **2: How could detect or filter out such requests to counter the attack. Could perturbation noise be added to mitigate the malicious impact of unlearning?**
>
> Thanks for the discussion on countering the attack detection. One strategy is to identify potential unlearning requests by tracking model parameter changes before and after unlearning. However, our experiments show that even with this method, our malicious unlearning attacks maintain high success rates. One reason is that these attacks are targeted at specific samples, not overall performance reduction. Another reason is that effective malicious data update requests cause slight model parameter changes, as observed. Another strategy involves using the maliciously unlearned model with other prediction models for ensemble-based methods. Yet, this requires additional models and learning costs. Also, recent studies highlight vulnerabilities in existing prediction models, making them susceptible to subtle attacks like data poisoning. Therefore, simultaneous attacks on multiple models could deceive ensemble systems.
>
> Additionally, in Section 9.1 of the Supplementary Material, we experimented with adding adversarial perturbation noises using adversarial training techniques to assess our attacks against robust deep learning models. Despite the success of adversarial training in enhancing adversarial robustness, our attacks remain potent against these robust models, as evident in Figure 1 and Figure 2 in Section 9.1 of the Supplementary Material. This further emphasizes the significance of our research, as no prior work has explored security risks introduced by selective forgetting.
>
> **3: Whether Anti-backdoor learning [r1] could be applied to mitigate the negative effect of proposed unlearning attack?**
>
> Thanks for your insightful question. We want to clarify that applying Anti-backdoor learning [r1] to mitigate our proposed unlearning attack is not feasible. Anti-backdoor learning assumes the presence of backdoored samples with triggers, focusing on trigger pattern identification and correlation removal. Our unlearning attacks do not involve the injection of triggers; we remove specific training samples rather than adding triggers. Furthermore, Anti-backdoor learning identifies low-loss backdoor examples early in training and removes correlations, while our attacks target malicious deletion during unlearning (instead of training). Consequently, the strategies of Anti-backdoor learning are unsuitable for mitigating our unlearning attacks.
>
> **4: Discussions on the potential limitations of this attack.**
>
> Due to space limitations in our main submission, we deferred the discussions on the potential limitations of our attack in Supplementary Material (see Section 2 in Supplementary Material). In the final version, we will incorporate the discussions on the potential limitations of our attacks from the Supplementary Material to the main manuscript.
>
> **5: Adding more convincing baseline comparisons in the experiments.**
>
> Thank you for your suggestions. We followed your advice and conducted experiments with two new baselines: input space clustering-based unlearning and representation space clustering-based unlearning. These clustering-based methods remove specific sample types, differing from the RandSearch baseline. In Table 1, we present the attack success rates of these clustering baselines on CIFAR-10 using first-order and second-order unlearning. We divided training data into 250 clusters using K-means for each baseline, based on input and representation space distances. Unlearning involved randomly selecting clusters and ensuring an equal number of unlearned samples as our methods. The results show that clustering-based unlearning baselines still perform poor low attack success rates. However, our proposed methods significantly outperform them by a large margin. This is because random cluster unlearning fails to guide importance scores for training samples to influence targeted loss in selective forgetting attacks.
>
> Table 1: Attack success rate of our proposed static forgetting attacks and new baselines.
> | Unlearning method  | RandSearch | Input space | Representation space | Ours |
> | :---: | :---: | :---: | :---: | :---: |
> | First-order | 0.08 ± 0.04 | 0.04 ± 0.03 | 0.14 ± 0.08 | 0.80 ± 0.04 |
> | Second-order | 0.10 ± 0.08 | 0.08 ± 0.05 | 0.12 ± 0.05 | 0.82 ± 0.04 |
>
> **Reference**
>
> [r1] “Anti-Backdoor Learning: Training Clean Models on Poisoned Data”, NeurIPS 2021.

---

> > ### Comment · Reviewer_MEi8 · 2023-08-19
> > **follow-up.**
> >
> > After reviewing the author's response, some of my concerns have been addressed. Currently, I believe that this work meets the standards for the conference and I am leaning towards accepting it.

---

> > > ### Author Response · Authors · 2023-08-19
> > > **Thank you for the post-rebuttal feedback**
> > >
> > > Dear Reviewer,
> > >
> > > Thank you very much for the appreciation of this work!
> > >
> > > Best regards,
> > > Authors of Paper5786

---

### Official Review · Reviewer_a1jN · 2023-07-07

**Soundness:** 3 good
**Presentation:** 3 good
**Contribution:** 4 excellent
**Rating:** 7
**Confidence:** 4

**Summary:**

This paper identified a novel class of machine learning attacks, i.e.,  ML models can be manipulated with malicious data update requests during the machine unlearning process.
The authors study two threat scenarios: (1)  selective forgetting attacks and (2) sequential selective forgetting attacks.
Specifically, in the static setting, an adversary can first select a subset of accessible training data and then launch the unlearning request with the selected subset in order to induce the victim model to become close to a target model.
In this setting, the authors achieve the attack goal by casting the discrete data selection process into a continuous optimization process.
On the other hand, in the sequential setting, data update requests occur sequentially and the adversary can modify any update request before it is received by the victim.
In this setting, the authors propose to train a policy network via reinforcement learning to output attack strategies according to the current environment state.
Comprehensive experiments verify the effectiveness of the proposed selective forgetting attacks.

**Strengths:**

1. This paper identifies a novel and important ML threat, selective forgetting attacks. Such a threat is realistic in the real world and therefore worth the ML security community to continue further research.

2. For the static version of the attack, how to efficiently select a subset to launch an attack is a challenging problem.
The authors address the problem by approximating the discrete indication function with a continuous function to perform GD-based optimization. This is a technically non-trivial solution.

3. The problem formalization for sequential attacks is comprehensive, as it considers (almost) all possible types of update requests("delete", "add", and "modify").

**Weaknesses:**

1. Two concerns about the sequential selective forgetting attacks:
    - In the black-box setting, According to this paper, the adversary could not access any training data. Therefore, to let the substitute model better imitate the behavior of the victim model (in order to train an effective policy network), the adversary may need to collect data that are similar to the real data. However, this could be difficult in the real world when the adversary does not have any prior knowledge about the private training data.

    - The adversary performs sequential attacks via modifying incoming update requests before they are received by the victim model, in which the modification strategy is by adding perturbations to the requested examples. This may result in a strange "paradox": if you modify the data that needs to be unlearned, is it still the original data? Wouldn't it result in the victim model "unlearning" something that it has never learned? Please comment.

2. Suggestion: including results of membership inference attacks (MIA) would be interesting. It would be interesting to see how selective forgetting attacks will affect the success rate of MIA.

**Questions:**

1. What is the usage of Theorem 1? It seems like removing Theorem 1 would not affect the conclusion of Section 3.1.

2. In Section 4, how to calculate the "attack success rate"? Is it the case that the authors first collect a set of data and then evaluate the victim's misclassification rate every time after processing an unlearn request?

3. The setting of selective forgetting attacks seems a little similar to that in [r1]. It would be interesting to compare [r1] with this paper.

**Reference**

[r1] Shumailov et al. "Manipulating SGD with Data Ordering Attacks". arXiv 2021.

**Limitations:**

The main limitation of this paper is that the threatening scenario of sequential attacks may not be realistic to some extent, which has been explained in Section "Weaknesses".
Nevertheless, I believe the merits of this paper suppress its disadvantages and would significantly benefit the ML security community.

---

> ### Author Rebuttal · Authors · 2023-08-09
>
> We really appreciate your thoughtful comments and valuable suggestions. Below, we provide our response to the questions and concerns.
>
> **1: Discussions on training the substitute model in the black-box setting where the adversary does not have prior knowledge about training data.**
>
> Thank you for addressing the black-box scenario, where the adversary lacks prior knowledge of private training data. In this setting, the adversary creates a synthetic dataset via existing model stealing attack techniques. They then train a local substitute model on this data, leveraging similar decision boundaries. Through traditional query methods, an ablation study was conducted on a three-layer ReLU neural network trained on MNIST as the target model. Table 1 indicates that the substitute model's performance approaches that of the target model more closely with an increasing number of queries. We also observed that their attack performance difference reduces as the query number increases. Query analyses will be included in the final version for this black-box setting.
>
> Table 1: Experiments on the discussed black-box setting.
> | Number of queries | Norm of parameter difference| Test accuracy difference |
> | :---: | :---: | :---: |
> | 2000 | 2.46 | 45.90% |
> | 4000 | 1.93 | 15.45% |
> | 6000 | 1.22| 9.68% |
> | 8000 | 0.67 | 5.85% |
> | 10000 | 0.19 | 2.51% |
>
> **2: Is it still the original data when modifying incoming update requests? Wouldn’t it result in the victim model “unlearning” something that it has never learned?.**
>
> Thank you for your insightful questions. The difference between modified and original data depends on update magnitudes. As we discussed in lines 216-217 of the main manuscript, for the setting of modifying incoming update requests, we are following existing partial data deletion works [r2, r3]. These methods involve perturbing requested examples [r2]. The adversary can exploit this and use the excuse of bad data quality issues (e.g., noises) or privacy to generate specific misclassifications.
>
> Regarding unlearning, it only affects acquired training knowledge. Unlearning techniques remove the influence of unlearning information from the pre-trained model. If the model never learned from the infomation, its influence is close to zero.
>
> **3: Including experiments on how forgetting attacks affect the success rate of membership inference attacks (MIA).**
>
> Thanks for highlighting this. Existing machine unlearning methods use MIA to measure the unlearning effectiveness. In contrast, our attack leverages existing unlearning techniques to induce misclassifications. Thus, MIA's success relies on how well unlearning methods remove maliciously requested unlearning samples.
> Following your suggestions, we evaluated membership inference attacks after maliciously unlearning certain CIFAR-10 training samples using various methods like first-order, second-order, unrolling SGD, amnesiac, and SISA. Based on [r4], we then compared these results with the baseline from the fully-trained original model. Table 2 displays the results. Notably, these unlearning methods offer similar performance deletion of malicious unlearning samples, as MIA metrics depend on the unlearning method's unlearning efficiency. To clarify, our work aims to study security vulnerabilities of the unlearning system by exploring the possibility of malicious data updates.
>
> Table 2: Experiments on membership inference attacks.
> | Model | Original | First-order | Second-order | Unrolling SGD | Amnesiac | SISA |
> | :---: | :---: | :---: | :---: | :---: | :---: | :---: |
> | Accuracy | 1.0 ± 0.0 | 0.52 ± 0.06 | 0.54 ± 0.06 | 0.55 ± 0.04 | 0.59 ± 0.05 | 0.48 ± 0.05 |
>
> **4: Discussions on Theorem 1 and the conclusion of Section 3.1.**
>
> Thanks for the question. In Theorem 1, we aim to study the difference between the original pre-trained model and its unlearned version (created after removing malicious unlearning samples). This helps understand the influence of these unlearning samples on the model's parameters. Notably, from this theorem, we can see that as more malicious unlearning samples are removed, the difference between the two models grows.
>
> **5: In Section 4, how to calculate the "attack success rate"? Is it the case that the authors first collect a set of data and then evaluate the victim's misclassification rate every time after processing an unlearn request?**
>
> In our experiments, we randomly sample a target class to attack as well as a set of targeted test samples in this class. We then run our selective forgetting attacks by optimizing which training data to unlearn to meet attack objectives (targeted or untargeted setting). After unlearning, we assess the misclassification rate on the targeted test samples. This process is repeated 10 times with varied random seeds, and we report the average results and standard errors.
>
> **6: Comparing the setting of the proposed selective forgetting attacks and that of [r1].**
>
> Thanks for highlighting the comparison with [r1]. Our method differs significantly from [r1]: First, [r1] targets training-time attacks, and assumes the adversary can alter the order of training data fed to the model. In contrast, our approach focuses on the unlearning process, and utilizes it to submit malicious update requests to achieve attack objectives.
> In addition, the setting of [r1] can be viewed as a special case of our sequential unlearning attacks, since we can use the defined data update requests (refer to Definition 2 in the main manuscript) to achieve the attack goal of [r1] via changing the order in which batches are supplied to the model during training.
>
> **Reference**
>
> [r1] "Manipulating SGD with Data Ordering Attacks". arXiv 2021.
>
> [r2] “Machine Unlearning of Features and Labels”, NDSS 2023.
>
> [r3] “Feature Unlearning for Generative Models via Implicit Feedback”, arXiv 2023.
>
> [r4] “Forgetting Outside the Box: Scrubbing Deep Networks of Information Accessible from Input-Output Observations”, ECCV 2020.

---

> > ### Comment · Reviewer_a1jN · 2023-08-19
> > **I vote for acceptance**
> >
> > Thanks to the authors for their response. All of my questions have been resolved. I believe this work will significantly contribute to the ML security community so I vote for acceptance.

---

> > > ### Author Response · Authors · 2023-08-19
> > > **Thank you**
> > >
> > > Dear Reviewer,
> > >
> > > We appreciate your encouraging response. We are delighted that our feedback has addressed your concerns.
> > >
> > > Thank you for your time again!
> > >
> > > Best regards,
> > > Authors of Paper5786

---

### Author Rebuttal · Authors · 2023-08-09

 We would like to express our great gratitude to all the reviewers for their valuable time, comments, and questions. We highly appreciate all the feedback and suggestions, which further help us improve our paper. We are greatly encouraged that they found our ideas and contributions to be novel and significant (Reviewers a1jN, MEi8, EC3c, and MTyE), comprehensive (Reviewers a1jN, MEi8, and EC3c), and technically sound (Reviewers a1jN, MEi8, and EC3c). We are grateful that they recognized the effectiveness of our methods (Reviewers a1jN, MEi8, EC3c, and MTyE), and the comprehensiveness of our experiments (Reviewers a1jN, MEi8, and EC3c).

We carefully considered all concerns and questions raised by the reviewers and provided our detailed responses to these concerns and questions raised in individual comments. We hope that our responses will address the concerns and questions provided by the reviewers. We look forward to participating in further discussions and answering any further questions posed by the reviewers.

Thank you for your time and consideration.

Best,

Authors of Paper5786

---

### Decision · Program_Chairs · 2023-09-21

**Decision:**

Accept (poster)

**Comment:**

This paper studies a threat model where an adversary attacks a machine learning system via malicious data update requests during the unlearning process. Unlearning is an interesting and new protocol in machine learning with practical applications. This paper makes a first attempt at evaluating potential and viability of malicious unlearning attacks. The proposed methods are technically sound and systematically evaluated. Additional results on higher resolution datasets added during the rebuttal further strengthen the empirical support.